# Hormonal Crossroads in Inborn Errors of the Metabolism Impact of Puberty and Dietary Interventions on Metabolic Health

**DOI:** 10.3390/metabo15040235

**Published:** 2025-03-28

**Authors:** Thomas Lundqvist, Rasmus Stenlid, Maria Halldin

**Affiliations:** 1Department of Women’s and Children’s Health, Karolinska Institutet, 171 76 Stockholm, Sweden; 2Unit for Pediatric Endocrinology and Metabolic Disorders, Karolinska University Hospital, 171 76 Stockholm, Sweden; 3Department of Medical Cell Biology, Uppsala University, 751 23 Uppsala, Sweden

**Keywords:** inborn errors of metabolism, obesity, puberty, treatment, sex hormones, metabolism, dietary intervention

## Abstract

**Background/Objectives**: Inborn errors of metabolism (IEMs) represent a diverse group of genetic disorders characterized by enzymatic defects that disrupt metabolic pathways, leading to toxic metabolite accumulation, deficits, or impaired macromolecule synthesis. While strict dietary interventions are critical for managing many of these conditions, hormonal and metabolic changes during puberty introduce new challenges. Advancements in early diagnosis and treatment have significantly extended the lifespan of individuals with IEMs. However, this increased longevity is associated with heightened risks of new medical problems, including obesity, insulin resistance, and type 2 diabetes mellitus (T2DM), as these complications share mechanistic features with those seen in obesity and T2DM. **Methods**: This mini-review examines current knowledge of the intricate interplay between pubertal hormones and metabolic pathways in IEM patients. **Results**: We address critical questions, such as if puberty intensifies the risk of metabolic derangements in these individuals and if there is a metabolic intersection where these disorders converge, leading to shared complications. We highlight the impact of puberty-induced hormonal fluctuations, such as growth hormone (GH) surges and sex steroid activity, on disorders like phenylketonuria, urea cycle defects, and fatty acid oxidation disorders. Moreover, we explore the role of dietary interventions in mitigating or exacerbating these effects, emphasizing the importance of balancing nutritional needs during growth spurts. **Conclusions**: A multidisciplinary approach integrating endocrinology, nutrition, and emerging therapies is advocated to optimize metabolic health during puberty. Addressing these challenges is critical for improving long-term outcomes for individuals with IEMs, particularly during this pivotal developmental phase.

## 1. Introduction

IEMs encompass a broad group of almost 1450 genetic disorders caused by defects in enzymes or transport proteins that disrupt essential metabolic pathways [1]. These disruptions can lead to toxic metabolite accumulation, substrate depletion, or impaired energy production, resulting in a wide range of clinical manifestations [2,3]. Depending on the specific enzymatic defect and its metabolic consequences, IEMs may present as acute, life-threatening metabolic crises or progress into chronic, multisystem disorders [4].

The area of inborn errors of metabolism is a quickly expanding field, not the least due to much improved genetic analysis. However, the global birth prevalence of all-cause inborn errors of metabolism (IEMs) is estimated to be 50.9 per 100,000 live births (95% CI: 43.4–58.4), with varying prevalence rates for different types of IEMs [5]. Amino acid disorders occur at an estimated rate of 14.7 per 100,000 live births, lysosomal storage disorders at 13.3 per 100,000, organic acidurias at 8.7 per 100,000, and mitochondrial disorders at 8.2 per 100,000. The prevalence can, however, differ significantly depending on factors such as population genetics, consanguinity rates, and the screening methods used in different regions [5].

One of the manifestations of IEM is impaired energy availability. It has been shown that many of the metabolic pathways that are often disrupted in IEMs are also dysfunctional in the development of insulin resistance and type 2 diabetes mellitus (T2DM) [6]. No genetic overlap between IEMs and T2DM has been identified to date; however, significant phenotypic similarities have been observed. Many of the biochemical changes seen in the prediabetic state, such as ectopic lipid storage, increases in acylcarnitines, and increases in branched-chain amino acids, are manifestations also seen in IEMs [6]. Chronic inflammation is widely believed to underlie the metabolic disturbances observed in the prediabetic and diabetic stages, with the accumulation of metabolic intermediates further exacerbating inflammatory processes [6]. Inflammation also plays a central role in many different types of IEMs, exemplified by Gaucher disease, a sphingolipidosis characterized by the accumulation of macromolecules [3,7,8]. Notably, inflammation serves as a common link between IEMs and T2DM and is prevalent in obesity [9,10], where it is a critical driver in the pathogenesis of obesity-induced T2DM [10].

Advances in newborn screening and early therapeutic interventions, such as dietary modifications, cofactor supplementation, and enzyme replacement therapy, have significantly enhanced survival rates in individuals with IEMs [11]. However, these advancements have also shifted the clinical focus toward the long-term management of complications arising from these conditions [12]. Despite the clear benefits of early detection and timely treatment, the clinical trajectory of IEMs remains highly variable, influenced by factors such as specific metabolic defects, adherence to treatment protocols, and the presence of coexisting complications [4].

The clinical manifestations of IEMs exhibit considerable variability, influenced by the specific enzyme or transport protein affected, residual enzymatic activity, and the metabolic demands at different life stages. These disorders may present as acute neonatal emergencies, episodic metabolic decompensations precipitated by stress or illness, or progressive developmental delays and multisystem organ dysfunction. Early and accurate diagnosis, often facilitated by newborn screening programs, is essential for implementing appropriate management strategies, including dietary modifications, cofactor supplementation, and enzyme replacement therapy. These interventions are critical for preventing irreversible damage and improving long-term prognoses.

As patients with IEMs transition from infancy to adolescence, the hormonal fluctuations associated with puberty present significant challenges. These changes frequently destabilize previously well-regulated metabolic pathways, adding complexity to treatment strategies. The metabolic demands of puberty, driven by increased energy requirements for growth and alterations in insulin sensitivity, often exacerbate pre-existing vulnerabilities in energy metabolism and skeletal development [13]. Moreover, puberty-induced hormonal surges, including elevated levels of GH, insulin-like growth factor-1 (IGF-1), and sex steroids, have profound and far-reaching effects on both metabolic and skeletal health [14,15].

This mini-review aims to examine the intricate relationship between puberty-induced hormonal fluctuations and disrupted metabolic pathways in individuals with IEMs, with a particular focus on disorders involving toxic intermediate accumulation and energy depletion. It explores how pubertal hormonal changes interact with disrupted metabolic pathways, amplifying the risk of further complications and potentially contributing to the development of metabolic syndrome over time. Additionally, this mini-review highlights the critical role of dietary interventions, emerging therapies, and a multidisciplinary approach in optimizing outcomes during this pivotal developmental phase.

## 2. Methods and Material

This mini-review was conducted by identifying peer-reviewed original research and review articles on inborn errors of metabolism (IEMs), puberty-related hormonal changes, insulin resistance, and targeted therapies. We focused on articles published within the last 10 years but also included earlier relevant articles. An electronic search was performed using PubMed, Web of Science, and Scopus, employing a variety of relevant keywords, including “Inborn Errors of Metabolism” and different diagnoses within the field of IEM such as “PKU”, “MSUD”, “puberty”, “hormones”, “insulin resistance”, “dietary interventions”, and “clinical trials.” The inclusion of multiple databases ensured a broader scope of research coverage and will thus maximize the identification of studies that are both high-quality and relevant. Bibliographies of key articles were reviewed to identify any relevant studies that may not have been found in the initial search, minimizing the potential for missing important research. While no formal systematic review protocol (such as PRISMA) was registered, this approach was intentionally chosen due to the exploratory nature of the mini-review, which aimed to synthesize a wide range of evidence rather than conducting an extensive systematic review. The focus was directed on selecting high-impact, peer-reviewed studies that directly address hormonal regulation, pubertal metabolism, IEMs, dietary management in IEMs, and broader endocrine or metabolic perspectives relevant to adolescents.

## 3. Results and Discussion

### 3.1. Classification of IEMs

IEMs are broadly categorized into three primary groups based on their underlying metabolic derangements. Each group reflects distinct pathophysiological mechanisms. By categorizing IEMs into these major groups, clinicians and researchers can better tailor the specific metabolic derangements, optimize treatment and anticipate complications, Table 1.

#### 3.1.1. Disorders of Intoxication

These conditions result from the toxic accumulation of metabolites due to defects in catabolic pathways. Toxicity is often triggered by dietary intake or physiological stress, which overwhelms the body’s capacity to process specific substrates.

Intoxication disorders include phenylketonuria (PKU), maple syrup urine disease (MSUD), and propionic and methylmalonic acidemias (PA and MMA). In PKU, mutations in the phenylalanine hydroxylase result in a deficiency in the enzyme phenylalanine hydroxylase, leading to a neurotoxic buildup of phenylalanine, which competitively inhibits the synthesis of catecholamines [16]. In MSUD, defective branched-chain ketoacid dehydrogenase (BCKD) activity causes an accumulation of branched-chain amino acids (BCAAs), particularly leucine, which could activate stress pathways like the mechanistic target of rapamycin (mTORC1) and impair insulin signaling [17]. MMA and PA are organic acidemias that result from enzyme deficiencies that block the metabolism of certain amino acids and odd-chain fatty acids. Toxic intermediates disrupt mitochondrial energy production and lead to oxidative stress. Patients may experience episodic metabolic crises, necessitating careful protein restriction and supplementation.

#### 3.1.2. Disorders of Energy Metabolism

Energy metabolism disorders involve defects in the production, storage, or utilization of ATP. Deficiencies exist in almost all enzymes involved in mitochondrial beta-oxidation. The most common disorder is medium-chain acyl-CoA dehydrogenase (MCAD) deficiency, which impairs fatty acid oxidation of medium-chain fatty acids, limiting the production of acetyl-CoA, a key substrate for the tricarboxylic acid (TCA) cycle. Deficiency in enzymes involved in beta-oxidation results in insufficient ATP production, manifesting as hypoglycemia and metabolic crises, particularly during fasting or illness [18,19].

#### 3.1.3. Disorders of Complex Molecule Metabolism

These disorders encompass abnormalities in the synthesis, degradation, or storage of macromolecules, often impacting multiple organ systems. For instance, Gaucher disease, a lysosomal storage disorder, arises from deficient glucocerebrosidase activity, resulting in the accumulation of glucocerebroside within macrophages. This pathological buildup leads to organomegaly and bone marrow suppression [7,20]. By categorizing IEMs into groups, clinicians and researchers can better tailor interventions to address specific metabolic derangements, optimize treatment outcomes, and anticipate complications.

### 3.2. Endocrinology in IEM

#### 3.2.1. Endocrine Disruptions in IEMs

IEM disorders have the potential to affect all endocrine glands. The most common endocrine manifestations in patients with IEMs include diabetes mellitus, thyroid dysfunction, and hypogonadism [21]. Some endocrinopathies observed in adults may result from an underlying metabolic disorder, particularly in cases with multisystemic involvement [21]. Despite this, the impact of IEMs on endocrine function remains poorly studied, highlighting the urgent need for further investigation [21]. Existing evidence also suggests that many endocrine manifestations may be subclinical, complicating their recognition and diagnosis.

Hormone production is an energy-intensive process reliant on adequate ATP generation and substrate availability [18,22]. The deviant metabolism in IEMs influences endocrine function in part through their direct impact on energy balance, hormone biosynthesis, and metabolic regulation. In patients with very long-chain acyl-CoA dehydrogenase deficiency (VLCAD), a disorder characterized by the defective metabolism of very long-chain fatty acids due to enzymatic deficiency [18], a small study reported low fasting glucagon concentrations [22]. This may compromise fasting glucose mobilization.

The dynamic interplay between metabolism and endocrinology becomes particularly pronounced during periods of increased energy demand, such as puberty, when hormonal fluctuations drive changes in substrate utilization, insulin sensitivity, and growth factor signaling. Hormonal and metabolic regulation is further modulated by paracrine and autocrine signaling, which integrate endocrine responses with cellular metabolism, energy production, neuroendocrine pathways, and immune function [21]. Additionally, the synthesis of peptide hormones, such as growth hormone (GH), insulin, and adrenocorticotropic hormone (ACTH), is contingent on sufficient amino acid availability. In phenylketonuria (PKU), where dietary protein intake is restricted, limiting essential amino acid availability, growth and endocrine regulation are adversely affected [23]. The accumulation of toxic metabolites may further exacerbate endocrine dysfunction. In a study of adult patients with PKU, Leal-Witt et al. reported that poor adherence to PKU treatment, leading to elevated plasma phenylalanine, was associated with altered insulin signaling, increasing the risk of glucose intolerance and insulin resistance [24].

Energy defects can affect all endocrine glands and may also result in deficiencies in cholesterol, a critical precursor for the synthesis of hormones such as cortisol, estrogen, and testosterone. These disruptions can impair normal puberty and the development of secondary sexual characteristics [19]. Patients with long-chain 3-hydroxyacyl-CoA dehydrogenase deficiency (LCHAD), a beta-oxidation defect, may develop hypoparathyroidism, while glycogen storage diseases (GSD) predominantly affect the pancreas and ovaries [21].

Defective glycosylation, as seen in congenital disorders of glycosylation (CDG), can lead to hypogonadism, a condition that can also be observed in cystinosis and iron overload [21]. Similarly, metal intoxications, such as those caused by hemochromatosis and Wilson’s disease, are linked to both hypogonadism and hypoparathyroidism [21]. Furthermore, deficiencies in lysosomal enzymes can impair growth and spermatogenesis, thus further complicating endocrine and reproductive health [8,25]. Mitochondrial respiratory chain defects and glycogen storage diseases (GSD) are among the IEMs most frequently associated with endocrine disturbances [21].

#### 3.2.2. Hormonal Modulation During Puberty in IEM Patients

Puberty is inherently associated with physiological insulin resistance, a mechanism thought to enhance glucose availability for anabolic processes such as growth. During this period, there is a marked increase in the secretion of GH, insulin-like growth factor-1 (IGF-1), follicle-stimulating hormone (FSH), luteinizing hormone (LH), and sex steroids, which collectively drive growth and maturation. However, these hormonal changes also influence insulin sensitivity, lipid metabolism, and energy homeostasis. GH and IGF-I can activate mTORC1, essential for cell growth and protein synthesis, but can also suppress autophagy. This effect may heighten oxidative stress and the risk of metabolic crisis in disorders with compromised mitochondrial or lysosomal function [26]. In addition, in patients with IEMs, these pubertal adaptations often exacerbate pre-existing metabolic imbalances [27]. Elevated GH levels, for instance, stimulate lipolysis, leading to increased circulating non-esterified fatty acids (NEFA), which impair insulin receptor function and further disrupt metabolic regulation [19], Table 2.

Growth and final height are frequently compromised in patients with IEMs. Growth retardation is a common feature, particularly in individuals with urea cycle disorders [28]. A Japanese study estimated that the final height of adult females with urea cycle disorders was approximately 10 cm shorter than the average height of their healthy counterparts [28]. Similarly, growth retardation during childhood is a hallmark of Gaucher disease type 1, where delayed puberty is also highly prevalent, particularly in individuals with more severe forms of the disease [14].

Puberty-related physiological insulin resistance is primarily driven by GH-induced lipolysis, which increases levels of non-esterified fatty acids (NEFAs) and impairs insulin signaling via serine phosphorylation of insulin receptor substrates (IRS) [19]. Estrogens, while improving insulin sensitivity by enhancing peripheral glucose uptake, also have a dual role in promoting fat deposition [29]. In the context of fatty acid oxidation defects, this dual effect can further disrupt lipid metabolism, increasing the risk of ectopic fat storage and chronic inflammation. Testosterone, on the other hand, enhances lean body mass and basal metabolic rate, thereby imposing greater energy demands. In metabolic disorders such as mitochondrial encephalomyopathies, these increased demands can overwhelm mitochondrial function, exacerbating energy deficits and oxidative stress.

Physiologically, insulin resistance typically resolves upon the completion of puberty. However, evidence suggests that in obese youths, insulin sensitivity often fails to normalize [27]. Additionally, children and adolescents with obesity exhibit higher insulin levels compared to their lean counterparts, regardless of glycemic tolerance [30,31]. The trajectory of insulin sensitivity during and after puberty in adolescents with IEMs remains largely unexplored, highlighting the need for further research in this area. Notably, hallmark dietary treatments for IEMs, characterized by high carbohydrate intake, protein restriction, and fasting avoidance, carry a recognized risk of promoting overweight or obesity [32,33,34].

The interplay between pubertal hormonal changes and pre-existing metabolic dysfunction can significantly amplify the risk of developing components of metabolic syndrome (MetS), including insulin resistance, hypertension, and dyslipidemia [35]. In children with obesity, elevated follicle-stimulating hormone (FSH) levels prior to puberty are associated with an increased likelihood of developing MetS during the pubertal transition [36]. Conversely, the demands of pubertal growth often necessitate a relaxation of dietary restrictions, particularly in disorders of protein metabolism, where higher protein intake becomes essential [23].

It is well established that women with galactosemia are at a high risk of developing primary ovarian insufficiency (POI) [37,38]. Most affected women require hormone replacement therapy (HRT), typically with estrogens, either to induce puberty or to maintain regular menstrual cycles, as ovarian function declines over time [38]. The underlying cause of POI in galactosemia remains unclear but may involve the toxic effects of endogenous galactose production and glycosylation abnormalities affecting FSH and LH. In contrast, gonadal function in males with galactosemia is typically unaffected [37]. Similarly, ovarian failure has been observed in some women with aminoacyl-tRNA synthetases 2 deficiency (ARS2 deficiency), a rare metabolic disorder that disrupts peptide synthesis [39].

Precocious puberty (PP) has been reported in patients with lysosomal storage disorders (LSDs). In a study of patients with Pompe disease undergoing enzyme replacement therapy, 55% exhibited PP, although the sample size was small (n = 9). Patients with PP also demonstrated significantly reduced predicted final height compared to those with normal pubertal timing [40]. Similarly, PP has been observed in individuals with Sanfilippo IIIA disease [41].

Impaired energy metabolism and chronic inflammation may influence bone health during puberty, a period in life when the building up of bone mineralization is crucial. Osteoblast and osteoclast function may be impaired, increasing the risk of osteoporosis and fractures in adulthood [13].

The mechanisms underlying these varied presentations underscore the delicate interplay between energy supply, hormonal demands, and potential toxicity from accumulated metabolites, especially during puberty.

### 3.3. Molecular Intersections in IEMs and Dietary Management

Dietary management is the cornerstone of treatment for IEMs, affecting the metabolism of small molecules and energy metabolism, aiming to minimize toxic metabolite accumulation while supporting growth and development and producing the required energy demand for everyday life. However, the complex interplay between amino acid metabolism, mitochondrial function, energy disruptions, and hormonal regulation during puberty creates unique challenges.

#### 3.3.1. Amino Acid Disorders and Insulin Signaling Pathways

##### PKU

In PKU, deficient activity of phenylalanine hydroxylase results in the accumulation of phenylalanine, which disrupts neurotransmitter synthesis and endothelial function. The low-protein, high-carbohydrate diet commonly prescribed for PKU patients can exacerbate hyperinsulinemia and insulin resistance, ultimately impairing glucose tolerance [42]. Additionally, this dietary pattern may reduce nitric oxide synthesis, increasing the risk of vascular dysfunction and hypertension [23,43].

The impact of PKU on body mass index (BMI) varies, with studies reporting either increased BMI or no significant difference compared to healthy controls [44,45,46]. Notably, research by Camatta et al. found that female adolescents with PKU exhibited a higher risk of elevated BMI and an increased percentage of body fat [46], Table 3.

##### MSUD

MSUD results from defects in the branched-chain alpha-ketoacid dehydrogenase complex, causing toxic BCAA accumulation. During puberty, increased IGF-1 and testosterone levels activate the mTORC1 pathway, further impairing insulin signaling and increasing metabolic stress by promoting the serine phosphorylation of IRS proteins, reducing their ability to activate downstream signaling through the phosphoinositide 3-kinase (PI3K)-AKT pathway. Pubertal hormones such as IGF-1 and testosterone further enhance mTORC1 activity, worsening insulin resistance and dyslipidemia [47]. The low-BCAA diet necessary to prevent neurological damage in MSUD further complicates metabolic control by limiting substrate availability for energy production. Balancing dietary protein to manage BCAAs without compromising growth remains a significant challenge [27], Table 3.

##### PA and MMA

PA and MMA are organic acidemias caused by deficiencies in propionyl-CoA carboxylase and methylmalonyl-CoA mutase, respectively. These enzymes play critical roles in the metabolism of the amino acids valine, isoleucine, methionine, and threonine and odd-chain fatty acids. Deficiency in these enzymes leads to the accumulation of propionyl-CoA and methylmalonyl-CoA, which inhibits enzymes in the TCA cycle, resulting in impaired ATP production and energy deficits [48].

Protein restriction is often used to limit the intake of precursor amino acids, but this approach carries a risk of growth delays. Furthermore, oxidative stress arising from mitochondrial dysfunction exacerbates insulin resistance, especially during puberty when anabolic demands are elevated [49].

Disruption of the TCA cycle and mitochondrial dysfunction also leads to increased production of reactive oxygen species (ROS), driving oxidative stress. ROS activate stress signaling pathways, such as c-Jun N-terminal kinase (JNK) and p38 mitogen-activated protein kinase (MAPK). These pathways phosphorylate insulin receptor substrate (IRS) proteins on serine residues, impairing insulin signaling and contributing to insulin resistance and glucose intolerance [49].

#### 3.3.2. Beta-Oxidation Defects

Beta-oxidation defects represent a broad category of disorders that impair energy production from fatty acids [4,19]. During beta-oxidation, acylcarnitines are metabolized through the beta-spiral in mitochondria, generating energy and progressively shortening acylcarnitines by two carbon atoms with each cycle [4,19]. Most fatty acid oxidation disorders require fat-restricted diets and avoidance of prolonged fasting to prevent metabolic decompensation. However, such dietary restrictions can impair steroidogenesis by limiting cholesterol availability, which may disrupt sex hormone production and pubertal development. Continuous nocturnal feeding, often prescribed to prevent fasting hypoglycemia, can further disrupt lipid metabolism and may contribute to an increased risk of metabolic syndrome over time [19]. Notably, the effects of continuous nocturnal feeding on metabolism and insulin sensitivity in IEMs remain largely unexplored yet represent a significant concern.

Lipid metabolism and glucose homeostasis are intricately connected [6]. Studies indicate that plasma acylcarnitine levels are elevated in individuals with obesity and diabetes, highlighting biochemical similarities between these conditions and IEMs [50,51,52]. Lipid metabolism in individuals with T2DM or obesity is profoundly altered, with increased lipid storage in organs such as the liver, muscle, and pancreas [6,53]. This suggests a potential intersection between these metabolic disturbances.

Acylcarnitines themselves have biological effects beyond energy metabolism. They can interact with cell membranes, potentially disrupting membrane integrity and function. Additionally, acylcarnitines have been shown to induce oxidative stress and promote a pro-inflammatory state, both of which are hallmark features of metabolic syndrome (MetS) [50,54,55].

Sirtuins, mitochondrial proteins involved in regulating gene expression and enzymatic activity, play key roles in oxidative stress responses, insulin resistance, respiratory function, and adipogenesis [56]. Of particular interest, SIRT3 modulates intermediary metabolites for energy production [57]. Sirtuins are implicated in the pathogenesis of insulin resistance and T2DM [56], but how their function is affected in IEMs remains unclear. Given their reliance on mitochondrial health, it is plausible that mitochondrial dysfunction in IEMs could significantly impair sirtuin activity.

The relationship between beta-oxidation defects and insulin sensitivity appears more complex than previously assumed. Studies in cell culture, animal models, and patients have reported both increased and decreased insulin sensitivity [6]. Unexpected results have been observed at the rate-limiting steps of beta-oxidation, specifically in the mitochondrial uptake of long-chain fatty acids via carnitine palmityoltransferase 1 (CPT1), carnitine palmityoltransferase 2 (CPT2), and carnitine-acylcarnitine translocase (CACT). For example, treatment with the CPT1 inhibitor Etomoxir in T2DM patients paradoxically improved insulin sensitivity, contrary to the hypothesis that lipid accumulation and lipotoxicity exacerbate insulin resistance [58]. On a cellular level, flux analysis in our group revealed diminished energy production in peripheral blood mononuclear cells (PBMCs) from patients with MCAD and VLCAD deficiencies, as evidenced by reduced oxygen consumption rates (OCR), compared to individuals with carnitine uptake disorders (CUD) who exhibited normal beta-oxidation profiles [18].

Given the recent paradigm shift in the treatment of non-syndromic obesity with GLP-1 receptor agonists, including in children [59,60,61], these drugs present a potential avenue for addressing metabolic alterations in children with IEMs. However, their application in this population would require careful consideration due to the unique clinical complexities of IEMs.

**Table 2 metabolites-15-00235-t002:** Molecular mechanisms of key hormones and their metabolic impact in inborn errors of metabolism patients.

Hormone	Normal Molecular Mechanisms and Metabolic Effects	Impact on IEM Patients
Growth Hormone (GH)	- Receptor Activation: GH binds GH receptor (GHR), activating JAK2/STAT5 signaling. - IGF-1 Synthesis: Stimulates hepatic IGF-1 production. - Lipolysis Promotion: Via hormone-sensitive lipase (HSL) phosphorylation, increases circulating free fatty acids (FFAs). - Insulin Antagonism: Reduces insulin receptor substrate (IRS) signaling, limiting GLUT4-mediated glucose uptake.	- Elevated FFAs: Increased FFA flux can overwhelm defective β-oxidation in mitochondrial disorders (e.g., VLCAD and LCHAD deficiencies). - Hyperglycemia Risk: GH-induced insulin resistance may exacerbate hyperglycemia in disorders with compromised gluconeogenesis (e.g., glycogen storage diseases). - Mitochondrial Overload: Excess FFAs burden dysfunctional mitochondria, raising oxidative stress and risk of metabolic crises.
Insulin-Like Growth Factor-1 (IGF-1)	- Receptor Binding: IGF-1 binds IGF1R (tyrosine kinase receptor). - PI3K/Akt/mTOR Pathway: Promotes mTORC1-mediated protein synthesis and cell growth. - MAPK/ERK Pathway: Further stimulates proliferation and anabolism. - Glucose Uptake: Enhances GLUT4 translocation and expression.	- Anabolic Accumulation: Heightened protein synthesis may accumulate toxic intermediates in amino acidopathies (e.g., PKU and MSUD). - Autophagy Inhibition: mTORC1 activation suppresses autophagy, which can worsen cellular damage in mitochondrial or lysosomal disorders. - Energy Imbalance: Increased anabolic drive can exacerbate ATP deficits in disorders of energy metabolism (e.g., MCAD deficiency).
Follicle-Stimulating Hormone (FSH)	- FSH Receptor (FSHR) Activation: Increases cAMP, activating PKA in ovarian granulosa or testicular Sertoli cells. - Steroidogenesis: Upregulates aromatase (CYP19A1) to convert androgens to estrogens. - Gametogenesis: Promotes follicle maturation in females and spermatogenesis in males.	- Steroid Hormone Deficiency: Defective cholesterol or precursor availability (e.g., fatty acid oxidation defects and certain organic acidemias) can reduce estrogen/testosterone synthesis. - Delayed Puberty: Chronic metabolic stress (e.g., in PKU with poor dietary control or GSDs with recurrent hypoglycemia) may delay the onset of puberty. - Fertility Concerns: May contribute to hypogonadism or subfertility in multisystemic IEMs (e.g., cystinosis and hemochromatosis).
Luteinizing Hormone (LH)	- LH Receptor (LHR) Activation: Additionally, increases cAMP/PKA signaling in ovarian theca cells and testicular Leydig cells. - Steroidogenesis: Stimulates enzymes (e.g., CYP11A1) for testosterone and estradiol production. - Ovulation Trigger: Mid-cycle LH surge induces follicular rupture.	- Disrupted Steroidogenesis: In energy metabolism defects, diminished ATP or substrate availability impairs sex steroid production. - Altered Lipid Handling: Changes in estrogen/testosterone levels can modify lipoprotein lipase (LPL) activity, influencing dyslipidemia risk. - Insulin Resistance: Elevated androgens (or abnormal estrogen) can alter adiponectin and promote insulin resistance, compounding metabolic risk in FAO disorders and certain organic acidemias.
Estrogen	- Estrogen Receptor (ER) Activation: ERα, ERβ nuclear receptors modulate gene transcription. - Metabolic Regulation: Enhances LDL receptor expression, upregulates fatty acid oxidation genes, and improves insulin sensitivity. - Anti-inflammatory: Inhibits NF-κB pathway, promoting an anti-inflammatory profile.	- Estrogen Deficiency: Insufficient steroid precursors (e.g., in severe FAO or GSD) can reduce estrogen levels, impairing bone health and pubertal progression. - Insulin Sensitivity Loss: When estrogen production is perturbed, beneficial effects on insulin signaling and lipid metabolism are diminished. - Increased CV Risk: Loss of estrogen’s anti-inflammatory influence can raise the risk of vascular complications (e.g., endothelial dysfunction in PKU).
Testosterone	- Androgen Receptor (AR) Activation: Regulates DNA transcription and promotes protein anabolism. - mTORC1 Signaling: Testosterone can increase mTORC1 activity, enhancing muscle protein synthesis. - Adipogenesis Modulation: Generally suppresses PPARγ activity, influencing fat cell differentiation. - Insulin Signaling: At moderate levels, can help maintain lean mass; excessively high levels can impair GLUT4 expression.	- Exacerbated Insulin Resistance: High testosterone can worsen IRS-1 dysfunction, amplifying hyperglycemia risks in conditions where insulin is already compromised. - Elevated Energy Demands: Increased muscle mass raises ATP requirements, potentially outstripping capacity in mitochondrial disorders (e.g., MELAS) or fatty acid oxidation defects. - mTORC1 Overdrive: Suppression of autophagy can be especially harmful in lysosomal storage disorders (e.g., Gaucher disease).

**Table 3 metabolites-15-00235-t003:** IEMs, affected pathways, pubertal hormone interactions, and clinical implications.

IEM Disorder	Affected Metabolic Pathway	Pubertal Hormone Interaction	Clinical Implications
Phenylketonuria (PKU)	- Deficiency in phenylalanine hydroxylase, causing phenylalanine accumulation and tyrosine depletion. - Neurotransmitter (dopamine, norepinephrine) synthesis is impaired.	- GH and Insulin Resistance: GH-induced lipolysis may exacerbate insulin resistance, especially under high-carbohydrate, low-protein PKU diets. - Estrogen and NO: Reduced precursor availability can diminish nitric oxide (NO) synthesis, impacting endothelial function.	- Cardiovascular Risk: Hypertension and endothelial dysfunction. - Neurological Issues: Persistent hyperphenylalaninemia can affect cognition and mood. - Risk of Excess Weight Gain: High-carb diet may promote hyperinsulinemia and adiposity, particularly in females.
Maple Syrup Urine Disease (MSUD)	- Defects in the branched-chain α-ketoacid dehydrogenase (BCKD) complex, causing buildup of BCAAs (leucine, isoleucine, and valine).	- mTORC1 Hyperactivation: Excess BCAAs can stimulate mTORC1, which is further upregulated by testosterone and IGF-1 in puberty. - Insulin Signaling Impairment: High BCAA levels can lead to serine phosphorylation of IRS-1, blunting downstream PI3K/Akt signaling.	- Metabolic Syndrome Risk: Enhanced risk of dyslipidemia and insulin resistance during adolescence. - Neurological Crises: Acute metabolic decompensation triggers neurological damage if BCAA levels spike. - Protein Balance Dilemma: Restrictive diets jeopardize growth.

### 3.4. Summary of Key Findings

Across multiple classes of IEMs, the physiologic stresses of puberty can amplify existing metabolic dysfunctions. Hormone production demands substantial energy and substrate availability, areas in which many patients with IEM disorders are already at a disadvantage. Strict dietary restrictions, while crucial for preventing toxic metabolite accumulation, may paradoxically promote overweight and/or insulin resistance. Weight reduction by reduced food intake could be dangerous with an underlying metabolic disorder. A recent shift towards a more aggressive pharmacological approach for pediatric obesity, including glucagon-like peptide 1 (GLP-1) receptor agonists, may hold promise for adolescents with IEM disorders but require careful observation for the unique metabolic risks.

A complicating factor is the interplay of inflammatory pathways and oxidative stress, both contributors to insulin resistance in conditions like T2DM. Patients with IEM disorders often share these pathophysiological hallmarks, raising concerns about an elevated risk for metabolic syndrome, compromised bone health, and reproductive endocrine disorders [10,13,36,37].

Overall, these findings underscore the need for carefully balanced and individualized dietary interventions and a multidisciplinary management approach to optimize growth, development, and fertility, as well as metabolic outcomes in adolescents with IEM disorders.

### 3.5. Implications for Drug Discovery and Clinical Trials

While dietary management remains the principal intervention for many IEMs, there is growing interest in pharmacological and biotechnological strategies that target specific metabolic nodes or inflammatory pathways. Puberty, as a critical window of metabolic and endocrine reprogramming, presents both an opportunity and a challenge for clinical trials:Hormone-targeted therapies are agents that modulate GH or sex steroids and could theoretically stabilize insulin sensitivity during puberty but require careful oversight to avoid adversely impacting growth or sexual maturation.GLP-1 receptor agonists are already employed in pediatric obesity, and drugs like exenatide or liraglutide (used off-label in certain contexts) show insulin-sensitizing and anti-inflammatory potential [9,59]. Small-scale studies in fatty acid oxidation or amino acid disorders may help define whether these can mitigate pubertal metabolic instability.Sirtuin modulators and gene therapy regulate mitochondrial function, oxidative stress, and insulin signaling [56]. Given that many IEMs involve mitochondrial deficits or chronic inflammation, sirtuin agonists or gene-based interventions could be promising, but robust protocols accounting for pubertal hormone dynamics.Genetic modulation and treatment with mRNA substrates are quickly developing new strategies in the treatment of IEM disorders.

Clinical trials must integrate the pubertal stage as a critical variable, given its large effects on insulin sensitivity, protein turnover, and hormone levels [15,26]. Strategies that combine gene therapy or enzyme replacement with puberty-specific considerations (e.g., ensuring adequate nutrient supply) can better delineate efficacy and safety profiles for this vulnerable group.

### 3.6. Future Directions

Despite extensive newborn screening and advancements in IEM treatment, the following research areas must be studied further:Longitudinal metabolic profiling: Prospective studies tracking adolescents through puberty to adulthood can clarify whether insulin resistance remits post-puberty or remains pathologically elevated in specific IEM cohorts.Personalized dietary interventions: The interplay between high-carbohydrate regimens, fasting avoidance, and growth demands requires nuanced protocols, particularly during growth spurts.Endocrine-targeted therapies: Understanding how subtle subclinical endocrine deficiencies (e.g., mild hypogonadism and suboptimal glucagon responses) influence long-term metabolic health may identify new intervention points.Biomarker discovery: Identifying inflammatory or lipid-related biomarkers (e.g., IL-18Rα, and specific acylcarnitines) can enable early detection of high-risk IEM adolescents who might benefit from additional interventions [9,50].

### 3.7. Additional Considerations: Age-Specific Interventions, Transition, and Collaboration

Although existing evidence underscores the general need for individualized dietary and therapeutic interventions, certain practical nuances can strengthen management strategies during puberty:

### 3.8. Age-Related Nutritional Guidance

Early adolescence (10–13 years): Rapid linear growth and fluctuating insulin sensitivity often necessitate slightly higher protein allocations or more frequent meals/snacks, depending on the underlying IEM. A modest protein increase can be considered in conditions like MSUD, balancing growth needs against the risk of elevated BCAAs.Mid-adolescence (14–16 years): Peak hormonal activity (particularly GH) can amplify lipolysis and insulin antagonism, warranting careful monitoring of blood glucose or ketone levels in fatty acid oxidation defects.Late adolescence (17–19 years): Insulin sensitivity may partially improve, but new social factors, such as sports participation and irregular sleep, may disrupt established feeding schedules. Flexible, nutrient-dense snack strategies can help prevent metabolic crises.

### 3.9. Transition into Adult Care

Many metabolic clinics transfer patients to adult services by 18–21 years of age. This handoff can jeopardize continuity if adult metabolic specialists have limited exposure to pediatric-onset conditions. Promoting a structured transition, where adolescents gradually take on responsibility for their nutritional management, helps maintain metabolic control beyond puberty [10,26].

## 4. Conclusions

In conclusion, the hormonal crossroads in inborn errors of metabolism (IEMs), particularly during puberty, present unique and complex challenges. The interplay between pubertal hormonal fluctuations and pre-existing metabolic imbalances can significantly exacerbate the risk of developing components of metabolic syndrome (MetS), including insulin resistance, hypertension, and dyslipidemia. While dietary management remains essential for preventing the accumulation of toxic metabolites, these interventions can inadvertently intensify metabolic disruptions during this critical developmental phase and later in life.

Advancements in treatment have allowed more individuals with fatty acid oxidation (FAO) disorders and other IEMs to reach adulthood, bringing the theoretical risks of insulin resistance, diabetes, and MetS into sharper focus. A deeper understanding of these hormonal and metabolic intersections is essential for developing targeted strategies to mitigate these risks and improve long-term patient outcomes. Addressing the hormonal crossroads inherent in IEMs, particularly during puberty, highlights the importance of a multidisciplinary approach that integrates endocrinology, nutrition, and emerging therapies to optimize metabolic health across the lifespan of affected individuals.

## Figures and Tables

**Table 1 metabolites-15-00235-t001:** Classification of inborn errors of metabolism.

Category	Characteristics	Examples of Disorders
Disorders of Intoxication	Toxic metabolite accumulation caused by defects in catabolic pathways. Often triggered by dietary intake or physiological stress.	Phenylketonuria (PKU), maple syrup urine disease (MSUD), propionic acidemia (PA), methylmalonic acidemia (MMA)
Disorders of Energy Metabolism	Impaired ATP production, storage, or utilization. May involve mitochondrial β-oxidation, oxidative phosphorylation, or glycogen breakdown.	Medium-chain Acyl-CoA dehydrogenase (MCAD) deficiency, very long-chain Acyl-CoA dehydrogenase (VLCAD) deficiency
Disorders of Complex Molecules	Abnormalities in synthesis, degradation, or storage of macromolecules affecting multiple organ systems (e.g., lysosomal or peroxisomal defects).	Gaucher disease, other lysosomal storage disorders (LSDs), glycogen storage diseases (GSDs)

## Data Availability

The data presented in this study are available on request from the corresponding author.

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
