# Peer review of "Hormonal Crossroads in Inborn Errors of the Metabolism Impact of Puberty and Dietary Interventions on Metabolic Health"

_metabolites, 2025, doi:10.3390/metabo15040235_

Round 1

Reviewer 1 Report

Comments and Suggestions for Authors

Thomas Lundqvist et al. submitted an interesting review about hormonal crossroads of IEM. The topic was of a certain significance, and might arouse some discussion in this field. The manuscript could be reconsidered by Metabolites after a Major Revision. Detailed comments:

  1. It would be better interpreted if the questions in the Abstract were rewritten in the form of declarative sentence.
  2. “Mini review” should be deleted in the Keywords.
  3. The title of the first section of manuscript must be Introduction.
  4. For the Classification of IEMs, it would be better to add a tabular or schematic summary for better understanding.
  5. Could the authors provide some epidemic statistics on the IEM?
  6. Prior to the Conclusion Section, it was suggested to add some discussion about the implications on drug discovery and clinical trials.
  7. It was found that the logical structure of the subtitles of sections was unclear. Please consider to name them numerically, like 1-1.1-1.1.1.

Author Response

  1. It would be better interpreted if the questions in the Abstract were rewritten in the form of declarative sentence.

The previous questions in the abstract have been rewritten in under the Method and Result    sections.

2. “Mini review” should be deleted in the Keywords.

The “ Mini review” has been deleted in the Keywords

3. The title of the first section of manuscript must be Introduction.

The first section has now the title “Introduction”

4. For the Classification of IEMs, it would be better to add a tabular or schematic summary for better understanding.

A new table (Table 1) has been included under the section 3.1 “Classification of IEM” line 144-145

5. Could the authors provide some epidemic statistics on the IEM?

We have provided some epidemic statistics line 43-52 in the manuscript

6. Prior to the Conclusion Section, it was suggested to add some discussion about the implications on drug discovery and clinical trials.

A section about drug and clinical trials implications has been added under section 3.5 “Implications for drug discovery and clinical trials “  has been added in the manuscript line  467-493.

7. It was found that the logical structure of the subtitles of sections was unclear. Please consider to name them numerically, like 1-1.1-1.1.1.

This has been done in the entire manuscript.

Reviewer 2 Report

Comments and Suggestions for Authors

The abstract effectively provides a concise summary of the study, highlighting the key issues related to inborn errors of metabolism (IEMs) and their interplay with puberty-induced hormonal changes. 

The introduction provides a strong foundation by contextualizing IEMs as a diverse group of genetic metabolic disorders. It emphasizes the impact of enzymatic defects on metabolic pathways and discusses how puberty exacerbates metabolic instability in affected individuals. 

As this is a mini-review, the methods section is somewhat integrated into the discussion rather than explicitly stated. The authors review existing literature on IEMs and their metabolic consequences, but there is no clear description of the criteria used for selecting studies, databases searched, or inclusion/exclusion criteria. Providing a structured methodology would enhance the rigor of the review. Additionally, specifying whether the review follows PRISMA guidelines or another systematic review framework could help validate the selection process.

The results are well-organized, with a detailed discussion on how puberty influences metabolic health in patients with IEMs.

The discussion is comprehensive and insightful, emphasizing the complexity of managing IEMs during puberty. However, while the results section presents a wealth of information, the connection between different subsections could be improved for better coherence. Additionally, the discussion would benefit from a clearer identification of gaps in current knowledge and suggestions for future research directions.

The conclusion effectively summarizes the key findings and emphasizes the need for a holistic approach in managing IEMs during puberty. It would also be helpful to include a stronger call for future research, particularly in areas such as individualized hormone-targeted therapies and the long-term metabolic consequences of different dietary interventions.

The reference list is extensive and up-to-date, citing relevant studies from peer-reviewed journals. 

Author Response

1. The abstract effectively provides a concise summary of the study, highlighting the key issues related to inborn errors of metabolism (IEMs) and their interplay with puberty-induced hormonal changes. 

The introduction provides a strong foundation by contextualizing IEMs as a diverse group of genetic metabolic disorders. It emphasizes the impact of enzymatic defects on metabolic pathways and discusses how puberty exacerbates metabolic instability in affected individuals. 

As this is a mini-review, the methods section is somewhat integrated into the discussion rather than explicitly stated. The authors review existing literature on IEMs and their metabolic consequences, but there is no clear description of the criteria used for selecting studies, databases searched, or inclusion/exclusion criteria. Providing a structured methodology would enhance the rigor of the review.

Additionally, specifying whether the review follows PRISMA guidelines or another systematic review framework could help validate the selection process.

We have added a Method and material section in the manuscript, section 2, line 115-136

2. The results are well-organized, with a detailed discussion on how puberty influences metabolic health in patients with IEMs.

The discussion is comprehensive and insightful, emphasizing the complexity of managing IEMs during puberty. However, while the results section presents a wealth of information, the connection between different subsections could be improved for better coherence. Additionally, the discussion would benefit from a clearer identification of gaps in current knowledge and suggestions for future research directions.

We agree to the comments. We have thus made changes in the manuscript for clarification.

We have named the different sections numerically, renamed the “Results” and “Discussion” sections as one section “Results and discussion” for the clarity, but organized the subsections numerically. We have moved text around for better coherence and tried to describe gaps of knowledge more clearly. We have added section 3.4 “ Summery of key findings” and added a section 3.5 about “Drug discovery and Clinical trials” and 3.6 “Future directions”, 3.7 “Additional considerations”, 3.8 “Age-related nutrional guidance” and 3.9 Transition into adult care”. A new table (Table 1) has been included under the section 3.1 “Classification of IEM” line 144-145.

With the overall changes made to the manuscript we are confident that this concern has been addressed.

3. The conclusion effectively summarizes the key findings and emphasizes the need for a holistic approach in managing IEMs during puberty. It would also be helpful to include a stronger call for future research, particularly in areas such as individualized hormone-targeted therapies and the long-term metabolic consequences of different dietary interventions.

This has been done, please see the answer above to the question

The reference list is extensive and up-to-date, citing relevant studies from peer-reviewed journals. 

Reviewer 3 Report

Comments and Suggestions for Authors

I appreciate the content, but I would like a better representation of it and would like to know if articles from the last 10 years have been included in this review, what are the criteria for inclusion, exclusion and how the articles were chosen. 

Please delete the empty spaces between paragraphs and move them closer together. 

Please add the chapter "material and method" respectively the chapter "results" and the chapter "discussion". Pay attention to the structure. 

Author Response

1. I appreciate the content, but I would like a better representation of it and would like to know if articles from the last 10 years have been included in this review, what are the criteria for inclusion, exclusion and how the articles were chosen. 

This has been described in a new section 2 “Methods and materials “ line 115-136

2. Please delete the empty spaces between paragraphs and move them closer together. 

This has been done

3. Please add the chapter "material and method" respectively the chapter "results" and the chapter "discussion". Pay attention to the structure. 

This has been done, please see above. The sections have been organized numerically for clarification.

Round 2

Reviewer 1 Report

Comments and Suggestions for Authors

Thanks for your revision.